# Engineering a Bi-Conical Microchip as Vascular Stenosis Model

**DOI:** 10.3390/mi10110790

**Published:** 2019-11-18

**Authors:** Yan Li, Jianchun Wang, Wei Wan, Chengmin Chen, Xueying Wang, Pei Zhao, Yanjin Hou, Hanmei Tian, Jianmei Wang, Krishnaswamy Nandakumar, Liqiu Wang

**Affiliations:** 1Energy Research Institute, Qilu University of Technology (Shandong Academy of Sciences), Jinan 250014, China; wangjc@sderi.cn (J.W.); wanw@sderi.cn (W.W.); chencm@sderi.cn (C.C.); zhaop@sderi.cn (P.Z.); houyj@sderi.cn (Y.H.); tianhm@sderi.cn (H.T.); wangjm@sderi.cn (J.W.); nandakumar@lsu.edu (K.N.); 2Key Laboratory of Interfacial Reaction & Sensing Analysis in Universities of Shandong, School of Chemistry and Chemical Engineering, University of Jinan, Jinan 250022, China; chm_wangxy@ujn.edu.cn; 3Cain Department of Chemical Engineering, Louisiana State University, Baton Rouge, LA 70803, USA; 4Department of Mechanical Engineering, The University of Hong Kong, Hong Kong

**Keywords:** blood-vessel-like, microchip, bi-conical, vascular stenosis, wall shear rate

## Abstract

Vascular stenosis is always associated with hemodynamic changes, especially shear stress alterations. Herein, bi-conical shaped microvessels were developed through flexibly and precisely controlled templated methods for hydrogel blood-vessel-like microchip. The blood-vessel-like microvessels demonstrated tunable dimensions, perfusable ability, and good cytocompatibility. The microchips showed blood-vessel-like lumens through fine embeddedness of human umbilical vein endothelial cells (HUVECs) on the interior surface of hydrogel microchannels, which closely reproduced the morphology and functions of human blood vessels. In the gradual narrowing region of bi-conical shape, fluid flow generated wall shear stress, which caused cell morphology variations. Wall shear rates at the gradual narrowing region were simulated by FLUENT software. The results showed that our microchannels qualified for performance as a vascular stenosis-like model in evaluating blood hydrodynamics. In general, our blood-vessel-on-a-chip could offer potential applications in the prevention, diagnosis, and therapy of arterial thrombosis.

## 1. Introduction

Cardiovascular, cerebrovascular, and peripheral artery diseases are potentially life-threatening risks that affect a significant percentage of the population. Atherosclerosis-associated vascular stenosis is one of the major pathological factors for disease onset [1,2], which often leads to abnormalities in local hemorheological properties [3]. The changes in blood hemodynamics, especially shear rate alterations, would, in turn, exacerbate vascular stenosis and might result in disorders, such as arterial thrombosis [4,5,6]. Unraveling the mechanism of how vascular stenosis interferes with local blood flow would have a positive impact on the prevention, diagnosis, and therapy of arterial thrombosis. 

Current researches on vascular stenosis are employing both in vivo animal models [7,8,9,10] and in vitro micro-engineered models [11,12,13]. Micro-engineered models, especially microfluidic organs-on-chips, could recapitulate aspects of the three-dimensional (3D) structure and microenvironment of human physiology with defined structure at miniature scale [14,15]. They provide an alternative in vitro platform for analyzing vascular thrombosis and are valuable complementary to the in vivo studies, especially for computational simulation of blood hemodynamics [13,16,17,18,19,20,21,22]. Most studies on artificial blood vessels utilize soft lithography in modeling microfluidic chips, as has been demonstrated in analyzing the effect of atherosclerotic geometries [13,16] and/or blood platelet function [23,24] on thrombosis formation, all focusing on illustrating the underlying mechanism. Despite the convenience in microchip construction, one drawback of soft lithography is that the cross-sections of the microchannels typically take the shape of a square or rectangle [13,16,24]. In a computational flow field study with such models, only the central portion of the microchannel could be approximately taken as parallel plate flow by overlooking the sidewall effect [23]. Therefore, micro-engineered channels with circular cross-sections and stenosis geometries would be preferable alternatives in modeling blood vessels for the in vitro analysis of arterial thrombosis.

In this study, we engineered gradual narrowing (or stenotic) blood-vessel-like microchips with bi-conical shape by using tapered capillary tubes as templates. The diameter and circular cavity shape of the generated microvessels could be well tuned by controlling the capillary templates. Gelatin was used to construct the microgel chip for its good cytocompatibility for cell adhesion and proliferation and excellent mechanical performance for structural replication. This hydrogel-based blood-vessel-like microchip has a smooth interior surface compared with those made with other methods, for example, 3D printing [21]. We also demonstrated that the microvessel was compatible with cell culture by embedding human umbilical vein endothelial cells (HUVECs) onto the lumen, which was followed by high-resolution confocal microscopy image analysis. Moreover, for proof-of-concept, Fluent software was employed to simulate the wall shear rates (WSR) for assessing the effect of vessel geometries on blood flow hydrodynamics.

## 2. Materials and Methods 

### 2.1. Materials

Cylindrical borosilicate glass capillary tubes (original inner diameters are 100 μm, 300 μm, and 500 μm, respectively) were purchased from World Precision Instruments (Johor, Malaysia). Gelatin and transglutaminase were purchased from Sigma (St. Louis, MO, USA). Fluorescein isothiocyanate dextran (FITC-dextran, average molecular weight MW = 40 kDa) and fluorescein isothiocyanate dextran 500,000-conjugate (FITC-dextran, average molecular MW = 500 kDa) were purchased from Sigma (St. Louis, MO, USA). HUVECs and RPMI1640 complete endothelial cell culture medium were purchased from CHI Scientific, Inc. (Maynard, MA, USA). Calcein-AM was purchased from Sigma–Aldrich (St. Louis, MO, USA). 4′,6-diamidino-2-phenylindole (DAPI) and Alexa Fluor 488 phalloidin were purchased from Life Technologies Corporation (Carlsbad, CA, USA). Penicillin-streptomycin solution was purchased from Hyclone (Chicago, IL, USA). Oil red was purchased from EKBAR. NP-40 lysis buffer was purchased from Beyotime Biotechnology (Shanghai, China). Human Von Willebrand Factor ELISA Kit was purchased from MSK Biotechnology (Wuhan, China). All chemicals and solvents were of the highest purity available and used as purchased without further purification. Distilled water used in the experiments was obtained from a Milli-Q-Plus water purification system (Millipore, Burlington, MA, USA).

### 2.2. Fabrication of Bi-Conical Blood-Vessel-Like Microchip

First, a plastic frame with holes on two sides was constructed to hold the capillary tube, and the diameter of the holes matched the outer diameter of the capillary tube (Appendix A). Second, the capillary tube was tapered using a micropipette puller (P-97, Sutter Instrument Co., Novato, CA, USA). The outer diameters of glass capillary tubes we chose were 318 μm, 576 μm, and 820 μm, respectively (original inner diameters of 100 μm, 300 μm, and 500μm, respectively). The capillaries were tapered at melting temperatures with different cycles (0, 5, 10, 15) to obtain different bi-conical shapes (i.e., different diameters). The outer diameters of the tubes were measured and recorded accordingly (Appendix A). 

Third, the capillary tube was horizontally positioned into the corresponding hole in the plastic frame and placed on a smooth and sterile glass sheet to ensure the bottom of the gelatin chip was flat. Gelatin (12.5%, w/w, 8 mL) solution was mixed with transglutaminase (10%, w/w,1 mL) and poured into the plastic frame; the crosslinking reaction was carried out under 37 °C for 2 h. After that, the capillary was removed, and the gelatin cavity was formed. The diameter of the gelatin cavity was determined by the outside diameter of the capillary. 

### 2.3. Characterization of the Morphology of Bi-Conical Blood-Vessel-Like Microchip and Investigation of Liquid Transport Properties

The shape and diameter of bi-conical blood-vessel-like microchips were visualized and measured under an inverted fluorescence microscope (Eclipse TS100, Nikon, Tokyo, Japan). 

To study the liquid transport properties of microvessels, low viscosity dimethyl silicone (viscosity coefficient 5 mPa∙s) was used as a blood analog fluid. Oil red O was added to give a red color. A peristaltic pump was applied to inject low viscosity dimethyl silicone into the microchannels to generate pulsatile flow [25]. A flexible pipe full of low viscosity dimethyl silicone was connected to both ends of the microchannels. The peristaltic pump worked at a constant rate of 60 cycles per minute, and a microscope was used to record the process.

### 2.4. Investigation of Microchannel Permeability

Permeability assay of the microchannels (channel of the microvessel) was investigated with 40 kDa FITC-dextran and 500 kDa FITC-dextran. FITC-dextran solutions were perfused into the channel of an artificial blood vessel by high-precision syringe pumps (LSP01-2A, LongerPump, Shanghai, China) at a constant rate of 100 mL∙min^−1^ to test the biomacromolecule permeability of the microchannels. The time slot was one and a half hours. A Leica DMi8 confocal microscope (Leica, Wetzlar, Germany) was used to record the whole process. The microchannel with the narrowest diameter of 125 μm was used as an example. 

### 2.5. Cell Culture

HUVECs were kept in RPMI1640 complete endothelial cell culture medium supplemented with 2 ng/mL vascular endothelial growth factor (VEGF) until 80% confluence (37 °C, 5% CO_2_). The cells were trypsinized and resuspended in complete medium. Cell density was calculated at 1 × 10^7^ cells mL^−1^ before use. 

### 2.6. Investigation of the Cytocompatibility and Endothelialization of Blood-Vessel-Like Microchip

Gelatin, a collagen hydrolyzate, has excellent properties for cell adhesion and proliferation. Gelatin microchannel does not require any special treatment and can be used directly for the culture of endothelial cells. HUVECs (1 × 10^7^ cells mL^−1^) were perfused into the channel and settled for 6 h to allow cells to attach to the interior surface. Then, the chip was flipped upside down, and HUVECs were perfused into the channel again to enable even attachment of HUVECs to the interior surface of the artificial blood vessel. The cells were cultured (37 °C, 5% CO_2_) for 4–5 days until a monolayer of cells covered all the interior surfaces of the microchannel and formed a round cross-sectioned chamber. Live/dead staining was performed using Calcein-AM and Propidium Iodide (PI) Double Stain Kit according to the manufacturer′s protocol to test the HUVECs survival rates. Alternatively, the HUVECs were stained with Alexa Fluor 488 and DAPI to visualize cell morphology. Confocal fluorescent microscopy was performed on a Leica DMi8 microscope with excitation lasers of wavelength 488 and 405 nm, respectively. 3D monolayer endothelium formed on the lumen of the artificial blood vessel was visualized by a confocal microscope at room temperature. The microchannel with the narrowest diameter of 390 μm was used as an example. 

### 2.7. Investigation of Cell Morphology and Alignment Changes at the Bi-Conical Region

To investigate the effect of shear stress on cell morphology and cell alignment at the bi-conical region, gelatin microvessels were endothelialized, as described. Then, a 1 mm thick polydimethylsiloxane (PDMS) block was placed at one end of the chip so that the height of one entrance of the microvessels was at the same level as the culture medium surface. Therefore, the flow would be formed by capillary force along the direction of the microvessels. After two days of incubation, cell morphology at and near the bi-conical region was visualized under an optical microscope. Cells that showed the shape of spindle instead of cobblestone and lining along the direction of fluid flow were taken as morphology variation and alignment changes. To quantify the morphologically changed cells, we defined the direction of fluid flow as 0°. All cells that took the shape of a spindle and their intersection angles with the direction of fluid flow were +30°~−30° and +150°~−150° and were counted as morphologically changed cells. 

### 2.8. Measurement of Endothelial Cell-Derived von Willebrand Factor

Bi-conical gelatin microchips cultured with HUVECs were prepared, as described above. Gelatin chips with straight vessels that of the original diameter (not pulled) and cultured with HUVECs were used as control. RPMI1640 complete medium (sterilized glycerol was added to adjust the viscosity at 5 mPa∙s) was perfused into the channel of artificial blood vessel by high-precision syringe pumps (LSP01-2A, LongerPump, Shanghai, China) at a constant flow rate of 1 mL·h^−1^ (0.32 cm·s^−1^) that is comparable to the blood flow rate of terminal arteriole (between arteriole 30 cm·s^−1^ and capillary vessel 0.03 cm·s^−1^) [26,27]. Chips were kept at 37 °C, and the perfusion lasted for 24 h. After that, HUVECs were collected and lysed by NP-40 lysis buffer. Endothelial cell-derived von Willebrand factor (vWF) was measured by Human von Willebrand factor ELISA Kit, following the manufacture’s protocol. A standard curve was drawn, and vWF was normalized to different cell numbers to optimize cell seeding numbers. The experiment was repeated for three times, and each vWF measurement was triplicated. 

### 2.9. Computational Simulation of Wall Shear Rates

FLUENT software (Version 16.1, Fluent Inc., Lebanon, NH, USA.) was used to predict the fluid flow profile and shear rate distribution inside the vessel geometries. The laminar flow module, which assumes incompressible flow and no turbulence, was used. The Gambit grid generator (Version 2.4.6, Fluent Inc., Lebanon, NH, USA.) was used to establish the flow field physical model and mesh-plot the vessel geometries. The total number of grids was 1 million to 3 million, depending on the diameter of the channels. An initial flow rate of 1 mL·h^−1^ was set on the inlet for the model. An entrance length of 1000 μm was chosen to provide a fully developed flow at the entry of the geometry. The inlet was set as a velocity-inlet, and the outlet was set as a pressure-outlet with no backflow. On the outer boundary of the geometries, excluding the inlets and outlets, a wall boundary condition was imposed. The number of iterations was set as 10,000. Data are presented as color distributions of WSR contours across the computational domain.

## 3. Results

### 3.1. Construction of Bi-Conical Vascular Hydrogel Models with Different Stenosis Structures

To fabricate the bi-conical shaped blood-vessel-like microchips, we chose three kinds of glass capillary tubes (original outer diameters of 318 μm, 576 μm, and 820 μm, respectively) and tapered at melting temperatures for 0, 5, 10, 15 cycles. The tapered bi-conical shapes with different diameters and the relationship between capillary deformation ratio and the number of cycles were provided in Appendix A. Then, the tapered tubes were used as templates for the construction of artificial blood vessels in hydrogel microchips. 

After tapering for five cycles, the narrowest diameters of the tubes were 125 μm, 330 μm, and 390 μm, with deformation length of 2970 μm, 3620 μm, and 3910 μm, respectively. To simplify, we took the sizes of capillary templates as the dimensions of microchannels in the subsequent analysis by ignoring minor deformation of microchannels when removing the templates. The tapered capillary templates (Figure 1A), engineered microchannels (Figure 1B), and cross-sections of microchannels (Figure 1C) at the narrowest point are illustrated in Figure 1 (capillary templates that were tapered for five cycles were used as examples). Here, we showed that the longitudinal shape of the microchannels was coaxial bi-conical, and the cross-section was circular. The interior surface of the microchannels was smooth (see cross-section picture in Figure 1). The gradual narrowing region of coaxial bi-conical shape was taken as a vascular stenosis-like model. 

### 3.2. Liquid Transport and Microchannel Permeability Assay of the Blood-Vessel-Like Microchip

Liquid transport is the primary function of the blood vessel. There should be no linkage or backflow in the process, and the wall of the blood vessel should be strong enough to withstand certain blood pressure. We injected low viscosity dimethyl silicone with oil red O as a red color into the microchannels to characterize liquid transport properties through a peristaltic pump. The fluid entered the microchannel from one end and flew out from the other end, with no linkage or backflow (Figure 2A), demonstrating excellent liquid transport ability. The elasticity of gelatin endowed the microchannels with good distensibility. Under the pumping pressure, the microchannels expanded and contracted regularly (Figure 2B–D, see also Appendix A), mimicking the pulsation of blood vessels and showing that the microchip could be taken as a simple blood vessel model.

Another important role of the blood vessel is the transportation of nutrition substances and exchange with tissues at the capillary vessel, to maintain the homeostasis of body fluid. Therefore, the permeability of the microchannel is an essential parameter in evaluating gelatin function as a suitable construction material. In this study, we used FITC-dextran with 40 kDa molecular and FITC-dextran with 500 kDa molecular weight to represent micro- and macromolecules in the blood, respectively. Figure 3 illustrates permeability assays of the microchannels on micro- and macromolecules (see also Appendix A). The micromolecules of FITC-dextran with 40 kDa molecular weight could traverse vessel wall and diffuse quickly, whereas the macromolecules of FITC-dextran with 500 kDa molecular weight could not traverse vessel wall and thus were retained in the channel. This was confirmed by fluorescence signals, where fluorescence signal of 40 kDa FITC-dextran decreased to the near background and fluorescence signal of 500 kDa FITC-dextran remained similar even after one and a half hours. This means the constructed microchannels were integrated and had different permeability to micro- and macromolecules. 

### 3.3. Cytocompatibility and Endothelialization of the Artificial Blood Vessel

Cytocompatibility is an essential indicator to evaluate the ability of a biomaterial to perform its desired function. Here, we performed Calcein-AM (Appendix A) and PI (Appendix A) double staining to detect cell survival rates after culturing for 4 days. The confocal microscopic images demonstrated that a monolayer of HUVECs was formed in the lumen of the artificial blood vessel, and most of the cells were alive (Figure 4A2–C2), and only a few were dead (Figure 4A3–C3). This was confirmed by 3D reconstruction of the endothelialized artificial blood vessel, and also the cross-section view took the shape of a circle (Figure 4A5–C6,B5–C6). Therefore, the blood-vessel-like microchip is highly cytocompatible and suitable for the in vitro reconstruction of the endothelial cell in vivo 3D microenvironment.

Similarly, endothelialization of the microvessel was visualized by F-actin (Appendix A) and DAPI (Appendix A) nucleic acid staining. Figure 5 demonstrates the confocal fluorescence microscopy images of the endothelialized microvessel in the deformation section of the microchannel. The bottom (Figure 5A), cross-section (Figure 5B), and top (Figure 5C) views of the microchannel showed that HUVECs formed a monolayer on the inner wall of the microvessel and that most HUVECs possessed typical cobblestone morphology. This was confirmed by 3D reconstruction of the endothelialized microvessel, and also the cross-section view took the shape of a circle (Figure 5A5–C5,A6–C6). 

### 3.4. Shear Stress Caused Cell Morphology and Alignment Changes at the Bi-Conical Region

Blood flow could generate shear stress on the inner wall of a blood vessel, especially at the stenosis region [28]. Shear stress would affect the cell morphology, as well as cell alignment lining within a microvessel. Cell morphology variations and cell alignment changes caused by flow-only shear stress around the bi-conical region is shown in Figure 6. It was obvious that the cells lining at the bi-conical region took the shape of a spindle instead of cobblestone, and the cells aligned along the flow direction. To quantify the cell morphology change, we divided the picture into four regions along the direction of the fluid flow with the same area. For the four regions, the spindle-shaped cell numbers are summarized in Table 1. Region A was at the entrance of the bi-conical region, and the spindle-shaped cell percentage was 21.46%. At the narrowest regions B and C, the spindle-shaped cell percentage was highest, showing that shear stress increased from A to C and was highest at B–C regions. Region D was the export of the bi-conical region, so the shear stress lowered down, and the spindle-shaped cell percentage was as low as 9.29%.

### 3.5. Measurement of Endothelial Cell-Derived vWF

Blood flow generated shear stress could cause endothelial cell secretion of vWF. In this study, the bi-conical vessel might exert more shear stress on the inner wall of the vessel than the straight vessel. The endothelial cells at the bi-conical microvessel with flow running for 24 h were designated as a test group, whereas the cells at the straight vessel with flow running for 24 h were designated as the control group. Untreated cells were used as a blank group. Figure 7 shows the ELISA measurement results. The value of vWF derived from the bi-conical vessel cells was 429.97 U/L, and the vWF value derived from the straight vessel cells was 169.40 U/L, while that of the blank group was 86.83 U/L. The results demonstrated that vWF derived from the bi-conical vessel cells was about three times higher than that from the straight vessel and was about five times the blank control. The vWF derived from the bi-conical vessel cells was significantly higher than the control group. Since vWF plays a critical role in thrombotic events [29], the evaluated vWF level would indicate an increased risk for the occurrence of thrombosis [30,31].

### 3.6. Computational Simulation of Hydrodynamics on the Blood-Vessel-Like Microchip Geometries

To test the feasibility of applying the microvessel geometries in analyzing hemodynamics and study the generated shear stress, FLUENT software was used to simulate the hydrodynamic changes in vessels of known size with stenosis structure. Three cases were analyzed, as shown in Table 2. Computational shear rate distributions and fluid flow profiles are illustrated in Figure 8. A comparison between different stenotic geometries (i.e., under different pathological conditions) showed a difference in the wall shear rates within and near the stenotic regions (Figure 8A–C). With the chosen volumetric flow rate, the shear rate distribution in the stenosis was similar; that is, the shear rate increased as the microchannel narrowed down. Typical wall shear rates in the stenotic sections of the narrowed vessel (case 1) were in the range of 1000 s^−1^, while they were in the range of 100 s^−1^ for the other two cases (cases 2 and 3). When the occlusion percentage increased to over 80% (Figure 8A), the highest shear rates were at 1600~1800 s^−1^. This means the higher the occlusion percentage, the higher the velocities and, consequently, shear rates.

In the experiment section, we did not apply a high velocity of flow because this might flush away the cultured cells at the interior surface of the microchannels. To compare with the real flow velocity, we simulated the shear stress distribution by using an initial flow rate of 4.83 × 10^−9^ m^3^ s^−1^ [21]. The results are depicted in Appendix A. For such a high flow velocity, typical wall shear rates in the stenotic sections of the narrowed vessel (case S1, seen in Appendix A) were in the range of 1000 s^−1^, while they were in the range of 100 s^−1^ for the other two cases (case S2 and S3, seen in Appendix A). For the occlusion percentage over 80% (Appendix A), the highest shear rates were at 14,000 s^−1^. Since shear rates higher than 5000 s^−1^ are considered pathological, and shear rates over 8000 s^−1^ could activate von Willebrand Factor [6,16], meaning that under physiological settings, the case S1 shall be at high risk.

## 4. Conclusions

In this study, we developed a bi-conical gelatin blood-vessel-like microchip by using a convenient, economical, and easy-to-manufacture method for vascular stenosis simulation by using a tapered capillary tube as a template. The gelatin chip showed good cytocompatibility and structural reconstitution performance. Permeability test, endothelialization of the microchannel, and formation of vascular cavity all demonstrated that the gelatin microchip could be viewed as a simple in vitro blood-vessel-like model. 

The shear stress at the bi-conical region increased as the vessel narrowed down and caused HUVECs morphology variations in this region. Cell morphology changes at the narrow regions might cause trauma on the lumen of the vessel, which might trigger plaque formation and further thrombosis formation. At the same time, shear stress is an important factor that might trigger physio-pathological alterations in the cells, especially von Willebrand factor activation [15,16] and subsequent platelet adhesion and aggregation [6,32]. To verify the effect of bi-conical shape on shear stress, endothelial cell-derived vWF was compared in both bi-conical and straight vessels. The results showed a 153.82% increase in vWF activation for the bi-conical vessel as compared to the straight vessel. To learn more about the shear stress distribution within the bi-conical vessel, we simulated the wall shear rates at the gradual narrowing region by FLUENT software. The simulation results showed that the bi-conical-shaped microchannels could be used as a vascular stenosis model in evaluating blood hydrodynamics. 

This micromodel could replicate the vascular microenvironment and intravascular hemodynamics, and it is expected to provide a new method for the in vitro analysis of atherosclerosis and hemodynamics. In the future study, by introducing smooth muscle cell and endothelial co-culture system, researchers could mimic the natural structure of blood vessels, which might be applicable in the research areas of in vitro tumor metastasis and angiogenesis.

## Figures and Tables

**Figure 1 micromachines-10-00790-f001:**
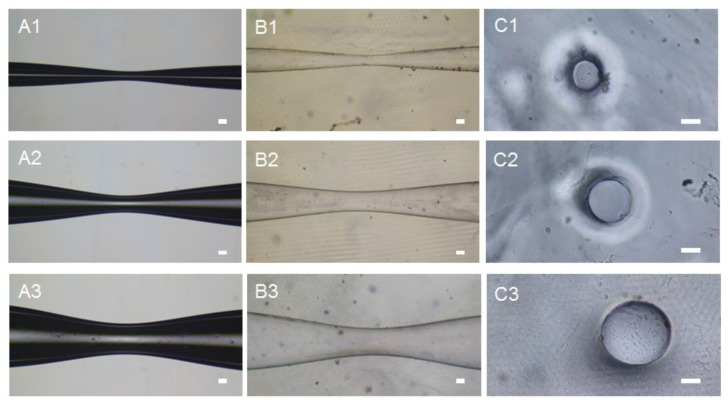
The tapered capillary tubes and engineered microchannels: (**A1**–**A3**) Photographs of the tapered capillary tubes; (**B1**–**B3**) Photographs of the gelatin chip with microchannels after removing the capillary templates and perfused with distilled water; (**C1**–**C3**) Cross-sectional view of the microchannels at the narrowest point. Scale bar, 100 μm.

**Figure 2 micromachines-10-00790-f002:**
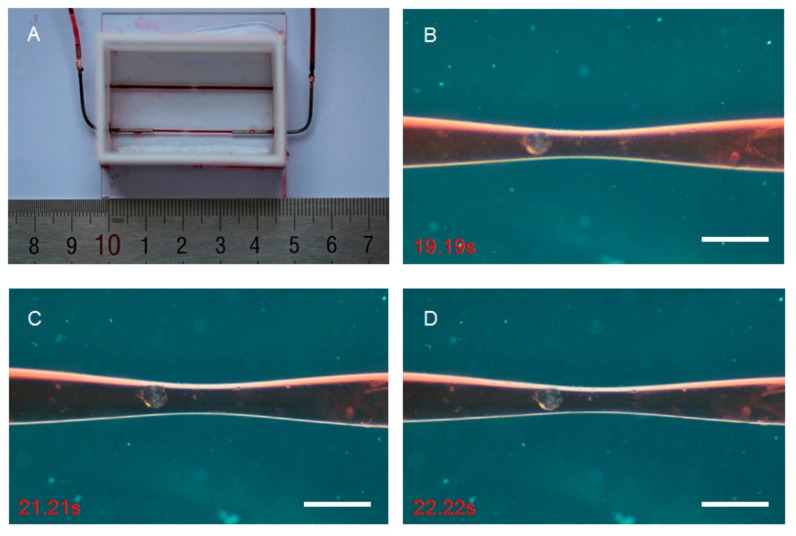
The microchip and liquid transport properties of artificial blood vessel: (**A**) The microchip perfused with oil red O, showing the liquid transport properties of the microchannel. (**B**–**D**) The original (**B**), expanded (**C**), and contracted (**D**) status of the microchannel. Scale bar, 100 μm.

**Figure 3 micromachines-10-00790-f003:**
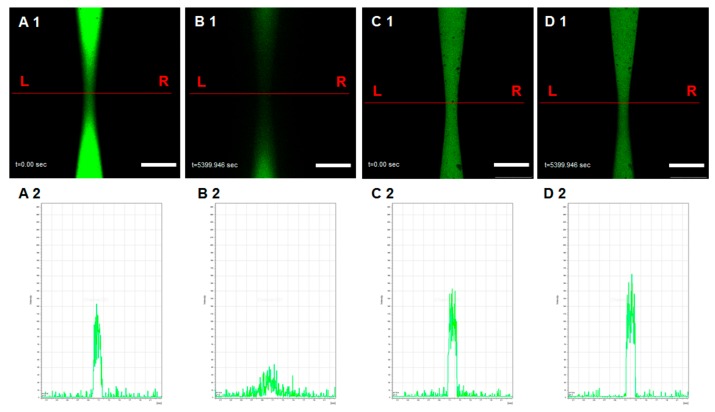
Microchannel permeability assay: (**A1**,**B1**) The permeability of microchannel for a micromolecule (A1,B1; FITC-Dextran, 40 kDa); (**C1**,**D1**) The permeability of microchannel for biomacromolecule (C1,D1; FITC-Dextran, 500 kDa). (**A2**–**D2**) indicate the fluorescence intensity. Scale bar: 500 μm.

**Figure 4 micromachines-10-00790-f004:**
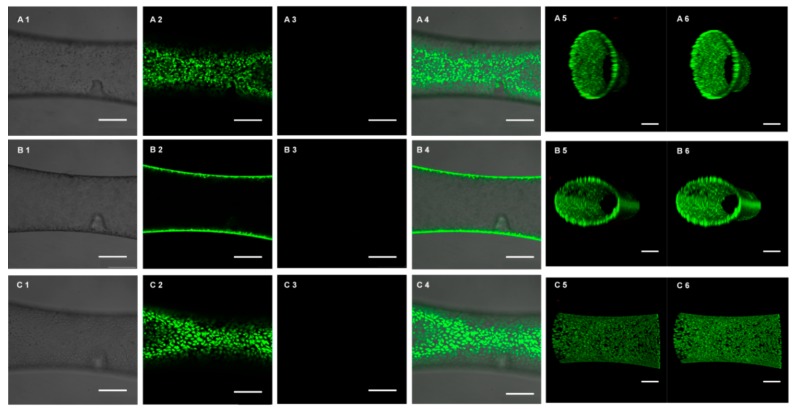
Calcein-AM and PI double staining, showing the endothelialized cell survival results of microvessels in the microfluidic chips. Confocal fluorescence microscopy images, showing the bottom (**A1**–**A4**), cross-section (**B1**–**B4**), and top (**C1**–**C4**) views of the microvessels in the deformation section of a channel with a smallest diameter of 390 μm. 1 for optical image; 2 for Calcein-AM staining; 3 for PI staining; 4 for merged image. Scale bar, 250 μm. 3D reconstruction views of the microvessels with HUVECs (A5–C5, A6–C6). (**A5**,**A6**) Top view of the microvessel with HUVECs; (**B5**,**B6**) One side angle view of the microvessel with HUVECs; (**C5**,**C6**) Side view of the microvessel with HUVECs. (A5–C5) stained by Alexa Fluor™ 488 Phalloidin and DAPI nucleic acid double staining, (A6–C6) stained by DAPI nucleic acid staining. Scale bar, 200 μm.

**Figure 5 micromachines-10-00790-f005:**
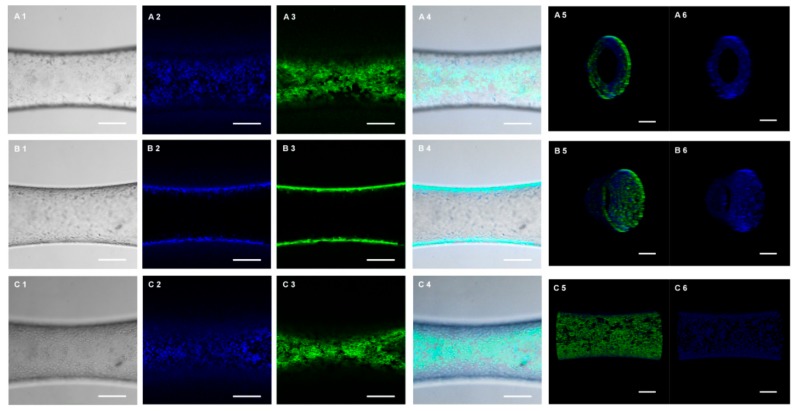
Cytoskeleton and nuclei double staining, showing the endothelialization of microvessel in the microfluidic chips. Confocal fluorescence microscopy images, showing the bottom (**A1**–**A4**), cross-section (**B1**–**B4**), and top (**C1**–**C4**) view of the microvessel within the deformation section of a channel with a smallest diameter of 390 μm. 1 for optical image; 2 for DAPI nucleic acid staining; 3 for Alexa Fluor™ 488 Phalloidin staining; 4 for merged image. Scale bar, 250 μm. 3D reconstruction views of the microvessel with HUVECs (A5–C5, A6–C6). (**A5**,**A6**) One side angle view of the microvessel with HUVECs; (**B5**,**B6**) Another side angle view of the microvessel with HUVECs; (**C5**,**C6**) Side view of the microvessel with HUVECs. (A5–C5) stained by Calcein-AM and PI double staining, (A6–C6) stained by Calcein-AM staining. Scale bar, 200 μm.

**Figure 6 micromachines-10-00790-f006:**
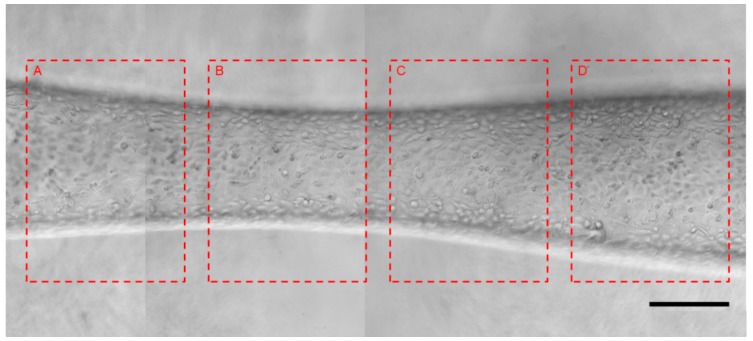
Shear stress caused cell morphology and alignment changes at the bi-conical region: From left to right, (**A**) Gradual narrowing channel region; (**B**) Narrowest channel region; (**C**) Gradual widening channel region; (**D**) Channel diameter widening to original size. Scale bar, 250 μm. Liquid flow direction, from A to D.

**Figure 7 micromachines-10-00790-f007:**
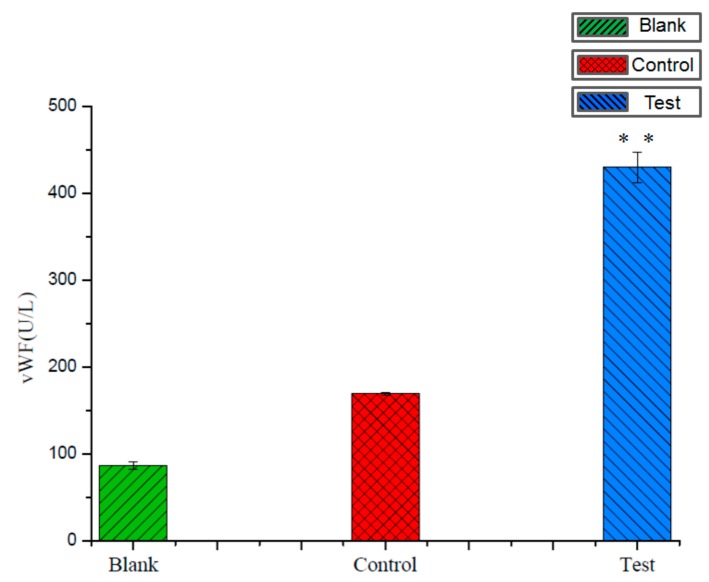
Cell-derived vWF (von Willebrand factor) activity assay: The cells at the bi-conical vessel with flow running for 24 h were designated as a test group, whereas the cells at the straight vessel with flow running for 24 h were designated as a control group. Untreated cells were used as a blank group.

**Figure 8 micromachines-10-00790-f008:**
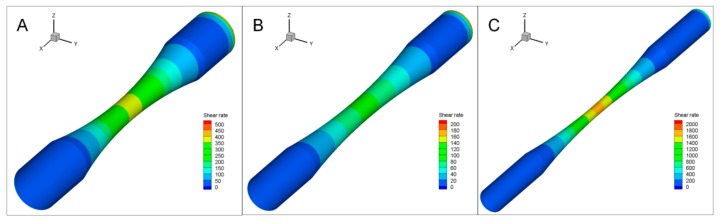
Computational simulation of wall shear rates (WSRs) in microchannels with a stenotic shape: Computational fluid dynamics analysis displayed WSR within the channel bottom in a microchannel. (**A**–**C**) Templates’ inner diameters were 390 μm, 330 μm, and 125 μm, respectively, and the stenosis percentage ranged from 67.2% to 84.5%.

**Table 1 micromachines-10-00790-t001:** Spindle-shaped cell number counts.

Region	Spindle-Shaped Cell Number	Total Cell Number	Spindle-Shaped Cell Percentage
A	38	177	21.46%
B	58	173	33.53%
C	56	181	30.94%
D	21	226	9.29%

**Table 2 micromachines-10-00790-t002:** Simulated Wall Shear Rates in Microchannels.

Case	Widest Diameter (μm)	Deformation Length (μm)	Narrowest Diameter (μm)	Percentage of Stenosis	Flow Rate	Widest Diameter (μm)
1	318	2970	125	84.5%	3.50 × 10^−3^	14,000
2	576	3620	330	67.2%	1.00 × 10^−3^	800
3	820	3910	390	77.4%	0.53 × 10^−3^	520

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
