# Peer review of "Engineering a Bi-Conical Microchip as Vascular Stenosis Model"

_micromachines, 2019, doi:10.3390/mi10110790_

Round 1

Reviewer 1 Report

The authors report on a simple stenosis geometry to micro-engineer channels as a bi-conical microchip model. The authors report on previous literature where similar models have been described, highlighting that the main focus of the current work is not to address the underlaying mechanism of stenosis, but more on the easiness of modelling a standard circular design with the possibility to achieve different diameters. This limits the application, representativeness and overall value of the proposed model. Physiological relevance and mechanistic understanding of stenosis would be more interesting for the development of the field and interest of the reader, rather than solely simulating perfect geometrical shapes.

More specifically, the authors should address the following suggestions to increase the relevance and quality of their current work:

1.       The study was performed using culture media and oil red, which is not representative of what the authors propose to model: blood flow. Considering this, the viscosity and cell content should be matched by perfusion of e.g. whole blood. Because the study was performed with non-physiologically relevant fluids, limited conclusions can be drawn for the effect of turbulence on constricted areas, how the cells attach to these decrease of diameter, and, overall, the effect of blood flow similarly to what occurs in disease conditions such as atherosclerosis.

2.       The authors should justify the use of gelatin for forming the vessels, as it has been reported as very unstable, leading to the collapse of formed structures within limited culture time frames such as 5-7 days. More information on the stiffness, smoothness and stability of the formed vessel obtained would add to the relevance of the model.

3.       The investigation of permeability was assessed on the vessels without cells. This is too simplistic to evaluate such a model, where the barrier function is critical. The comparison of permeability profiles need to be performed in the presence and absence of the cells, throughout the defined culture times.

4.       The cell culture section is described very poorly, not even including the definition of the culture medium components, supplier, concentration of growth factors, etc.

5.       Regarding the investigation of the biocompatibility, in section 2.6, this term should be rephrased to cytocompatibility, as none of the verification was performed in vivo. Also, the HUVEC cell seeding density is repeated unnecessarily in line 115 and again in 116.

6.       In section 2.7 the authors describe the process of evaluated the spindle shape of the cells, yet fail to assess the overall cell alignment, which will be clearly affected by the shear.

7.       In section 2.8 the authors report on a co-culture with HUVECs (line 137); this is not correct. Only one cell type is used. Moreover, the vWF was not normalized to total cell number or DNA content, hampering any conclusion derived from this data. In this same section please rephrase the last sentence, including defining triplicated.

8.       In the results section 3.1, the authors state that minor deformations in the micro-channels were ignored (line 169). These deformations should not be ignored, as it affects the smoothness of the channels, therefore interfering flow and cell shape/orientation.

9.       Table 1 should state more clearly to what A, B, C and D refers to. Is this related to figure 6? The limited description in the table caption leads to confusion of the reader.

10.   In section 3.2, the authors describe the system as mimicking the pulsation of real bool vessels (line 189). This is a rather bold statement regarding the limited experimental data present in the paper, not including blood in the perfusion of the model or variation in fluid viscosity. Also no data is acquired under pulsatile flow. The staining solution used, Oil red O, has no clear similarities with whole bool, except for the red color, which limits its relevance.

11.   In section 3.3 the authors emphasize on biocompatibility, please rephrase according to point 5 above. To claim suitability of the model, experiments need to be carried using whole blood, and describing the kinetics of platelet attachment and activation by collagen (which formed this model’s vessels).

12.   In section 3.6, the sentence using the term “planted cells” needs rephrasing to adequate biological scientific terms.

13.   In the conclusions, it states that the current gelatin chip “has excellent performance in cell adhesion/proliferation”. The data to justify this statement is not present in the current paper, as there’s neither adhesion efficiency studies nor assessment of cell proliferation profiles.

14.   Overall, several spelling mistakes are present throughout the text, as well as unsuitable sentence constructions. Examples include: “preperable” (line51) instead of preferable, “descibed” (line 137) instead of described, “protocal” (145) instead of protocol. This can be easily solved by any spelling check and it affects the overall reading experience. Sentence construction inappropriateness includes the example: “Quickly mix gelatin …”(line 88), which is written as lab protocol/SOP and not suitable for the M&M section of a paper. Last sentence of conclusion need serious rephrasing. As advice, I would recommend to perform a double check with a native English speaker/writer before submission.

Reviewer 2 Report

This is a very well presented paper on the microfabrication of a more realistic vascular stenosis model. Overall it's a nice and sound paper; not ground breaking but still nice scientific work. My only suggestion would be to improve the image quality of Fogs 4 and 5, especially the dark parts of them, since they are really illegible when printed. Other than that I think it could be published as is.

Round 2

Reviewer 1 Report

The authors have addressed all suggestions and modified the manuscript accordingly. No further comments regarding the modified version. The overall quality of the manuscript has increased and is now sufficient for publication.